# β-catenin mediates growth defects induced by centrosome loss in a subset of APC mutant colorectal cancer independently of p53

**Mohamed Bourmoum**[1], **Nikolina Radulovich**[2], **Amit Sharma**[1], **Johnny M. Tkach**[1], **Ming-Sound Tsao**[2], **Laurence Pelletier**[1,3]*

**1** Lunenfeld-Tanenbaum Research Institute, Sinai Health System, Toronto, ON, Canada, **2** University Health Network, Ontario Cancer Institute/Princess Margaret Cancer Centre, Toronto, ON, Canada, **3** Department of Molecular Genetics, University of Toronto, Toronto, ON, Canada

* pelletier@lunenfeld.ca

**Data Availability Statement:** All relevant data are within the manuscript and its Supporting Information files.

## Abstract

Colorectal cancer is the third most common cancer and the second leading cause of cancer-related deaths worldwide. The centrosome is the main microtubule-organizing center in animal cells and centrosome amplification is a hallmark of cancer cells. To investigate the importance of centrosomes in colorectal cancer, we induced centrosome loss in normal and cancer human-derived colorectal organoids using centrinone B, a Polo-like kinase 4 (Plk4) inhibitor. We show that centrosome loss represses human normal colorectal organoid growth in a p53-dependent manner in accordance with previous studies in cell models. However, cancer colorectal organoid lines exhibited different sensitivities to centrosome loss independently of p53. Centrinone-induced cancer organoid growth defect/death positively correlated with a loss of function mutation in the APC gene, suggesting a causal role of the hyperactive WNT pathway. Consistent with this notion, β-catenin inhibition using XAV939 or ICG-001 partially prevented centrinone-induced death and rescued the growth two APC-mutant organoid lines tested. Our study reveals a novel role for canonical WNT signaling in regulating centrosome loss-induced growth defect/death in a subset of APC-mutant colorectal cancer independently of the classical p53 pathway.

## Introduction

Colorectal cancer (CRC) is the third most common and the second most deadly cancer type worldwide (almost 2 million cases and 1 million deaths in 2020) [1]. CRC is a highly heterogeneous disease. Large-scale transcriptomic analyses led to CRC classification into four consensus molecular subtypes (CMS): CMS1 (MSI Immune, 14%), hypermutated, microsatellite unstable, strong immune activation; CMS2 (Canonical, 37%), epithelial, chromosomally unstable, pronounced WNT and MYC signaling activation; CMS3 (Metabolic, 13%), epithelial, evident metabolic dysregulation; and CMS4 (Mesenchymal, 23%), marked TGF-β activation, stromal invasion, and angiogenesis [2]. Mutations affecting key genes regulating cell proliferation, differentiation, and death accumulate in neoplastic cells, giving them a survival advantage

**Funding:** This work was funded by grants from the Krembil Foundation, the CCSRI and Compute Canada RAC to LP and a CIHR Fellowship (MFE 187836) and Hold'em For Life Oncology Fellowship to AS. The Network Biology Collaborative Centre at the LTRI is supported by the Canada Foundation for Innovation, the Ontario Government, and Genome Canada and Ontario Genomics (OGI-139). The funders had no role in study design, data collection and analysis, decision to publish, or preparation of the manuscript.

**Competing interests:** The authors have declared that no competing interests exist.

over surrounding normal intestinal epithelial cells [3]. These mutated genes lead to abnormal expansion of premalignant tissue into adenomas that have the potential to completely transform into invasive carcinomas due to additional genetic aberrations [3, 4]. Alterations in certain genes, such as APC and KRAS, have been shown to occur early whereas other genetic events are typically only observed in more advanced disease states, like the loss of function p53 mutations [3, 5].

The centrosome is the major microtubule-organizing centre (MTOC) in eukaryotic cells, consisting of two centrioles surrounded by an electron-dense matrix, the pericentriolar material (PCM). Centrosomes regulate many cellular processes including cell motility, adhesion, and polarity in interphase, and organizing the spindle assembly during mitosis [6]. In addition, in quiescent cells, centrioles can act as basal bodies that anchor cilia and flagella, which play important roles in physiology, development and disease [7]. Numerical and structural centrosome aberrations are frequent in many cancers and can be associated with genomic instability. Whether changes in centrosome numbers are a cause or consequence of oncogenic transformation remains a matter of debate [8] but in certain cancer types, evidence suggests that centrosomal defects might occur very early in tumorigenesis [9]. Centriole duplication is tightly controlled so that cells have precisely two centrosomes [10]. The serine-threonine protein kinase Polo-like kinase 4 (Plk4) plays a pivotal role in this process [11]. Wong et al. [12] developed centrinone B, a selective Plk4 inhibitor and found that centrinone-induced centrosome loss irreversibly arrested normal cells in a senescence-like G1 state by a p53-dependent mechanism whereas p53-mutant cancer cells could proliferate indefinitely after centrosome loss.

The development of human organoids has greatly benefited the biomedical research community as they filled the gap between the animal models, lacking human specificity, and the 2D cell models, lacking biological complexity. Since their establishment, human colorectal organoids [13] have been widely used to address different biological questions and to model colorectal cancer.

Here, using centrinone B, we sought to determine the role of centrosomes in the growth and survival of normal and cancer human-derived colorectal organoids (HCOs).

## Results

### Centrosome loss represses human normal colon organoid growth in a p53-dependent manner

We first assessed the effect of centrosome depletion on normal HCOs growth using centrinone B. Normal colon organoids were grown from single adult intestinal stem cells in the presence of DMSO or centrinone B. Centrinone B treatment strongly repressed the growth of the human normal organoids measured by the relative average organoid area (Fig 1A and 1B) (~88% reduction of the average organoid area). Previous studies have shown that centrosome loss irreversibly arrested normal cells in a senescence-like G1 state by a p53-dependent mechanism [12]. To test this in our model, we generated a p53 knockout human colon organoid line by infecting adult intestinal stem cells with an inducible lentiviral system expressing CRISPR/Cas9 and a p53 gRNA. *TP53*[-/-] organoids were enriched by Nutlin-3a selection for two weeks as described previously [14]. Short insertions and deletions (INDELs) at the p53 gene locus were confirmed using CRISPR TIDE analysis [15] (Fig 1C) and the loss of p53 protein was confirmed by immunofluorescence (Fig 1D) and Western blotting (Fig 1E and 1F). We noticed that the area of *TP53*[-/-] normal organoids was approximately 55% greater than wild type organoids (Fig 1B). More interestingly, we found that p53 depletion partially rescued the human organoid growth in the presence of centrinone B (~55% decrease in the average organoid area after centrinone B treatment in *TP53*[-/-] organoids versus 88% decrease in wild-type

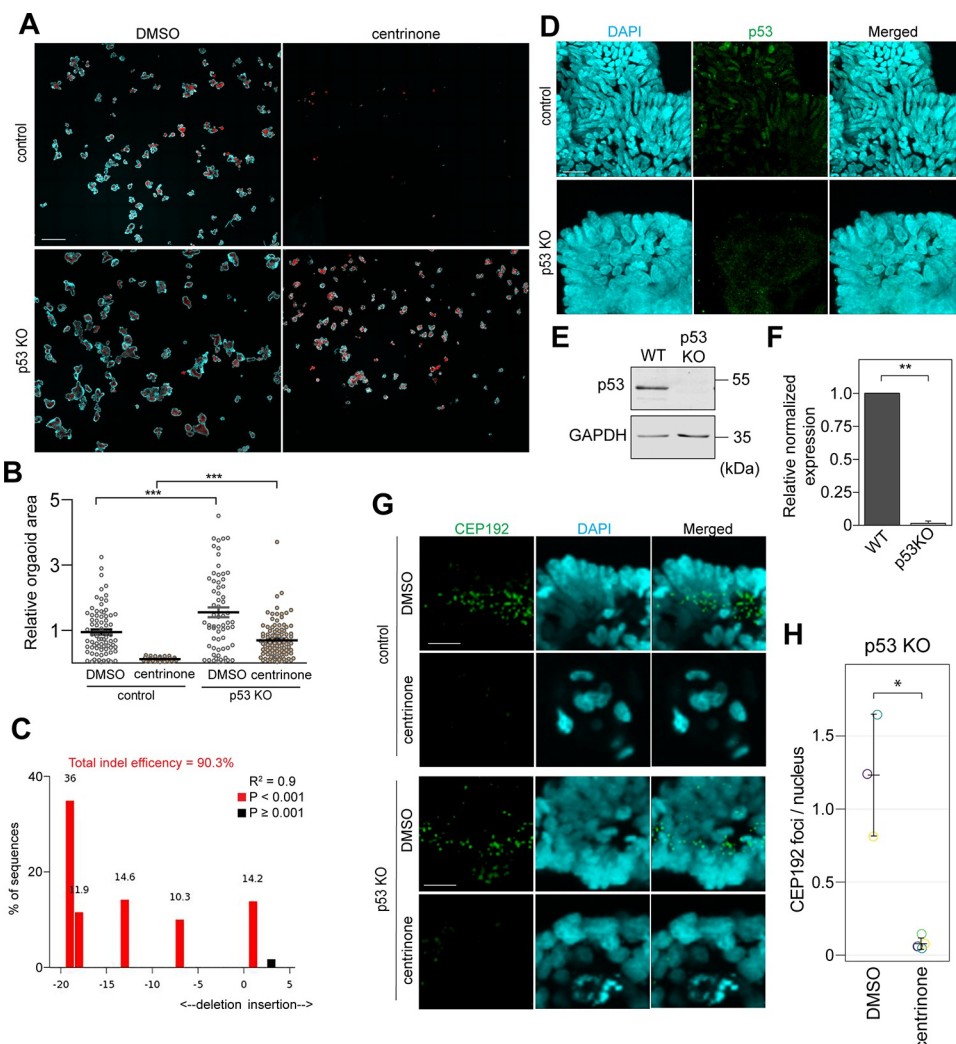

**Fig 1. Centrosome loss represses normal HCOs growth in a p53-dependent manner. A)** Control and p53 KO normal HCOs were grown from single adult stem cells in the presence of DMSO or 0.5 μM centrinone B for 14 days. Organoids were then fixed, stained with DAPI to label DNA (blue) and phalloidin to label actin (red) and imaged. Merged maximum intensity projections of images are shown. **B)** The areas of individual organoids were quantified in the merged maximum intensity projections (MIPs) of images from (A) (n = 3) and presented in the graph as values relative to the DMSO control where every dot represents an organoid (***$P<0.001$, One-way ANOVA with Bonferroni post-hoc). **C)** Tide analysis of p53 gene indel efficiency in the normal HCO p53 KO line. The algorithm provides the $R^2$ value as a goodness-of-fit measure and calculates the statistical significance for each indel. Red represents significant indels ($P<0.001$) Note that the total indel efficiency is ~90.3%. **D)** Representative maximum intensity projections of p53 immunofluorescence staining in control and p53 KO normal HCOs. **E)** Normal human organoids were extracted from Matrigel and lysed. p53 and the loading control (GAPDH) protein levels were assessed using western blot analysis. **F)** Quantification of (E). Protein levels from three independent experiments were quantified and presented in the graph (n = 3, **$P<0.01$). **G)** High-resolution maximum intensity projection images of CEP192 (centrosome) and DNA (DAPI) immunofluorescence staining in selected normal HCOs from (A). **H)** The number of CEP192 foci and nuclear objects was assessed in independent z-sections spanning three (DMSO) or five (centrinone) independent organoids. The foci to nucleus ratio was determined for each organoid (*$P<0.05$). Scale bars are (A) 500 μm, (D) 25 μm and (G) 10 μm.

organoids) (Fig 1A and 1B) indicating that the centrinone-induced growth defect in human colon organoid is p53-dependent in accordance with previous reports in tissue culture cells [12]. In wild-type and $TP53^{-/-}$ human normal colon organoids, centrinone-induced centrosome loss was confirmed by staining for the centrosomal protein CEP192 (Fig 1H and 1G).

## Patient-derived cancer colorectal organoids exhibit different sensitivity to centrosome loss independently of p53

To determine whether centrosomes are essential for colorectal cancer growth, we next assessed the effect of centrosome depletion on patient-derived cancer HCOs growth derived from three independent different patients (CSC-406, POP-092 and POP-112, see methods). In contrast to previous studies in cell models [12], we observed that the three cancer colorectal organoid lines exhibited different sensitivity to centrosome loss independently of p53 (Fig 2A and 2B). Whole exome sequencing data revealed that each colorectal cancer lines harboured a missense p53 mutation (S240R in CSC-406, R248W in POP-092 and R248Q in POP-112) (S1 File, Fig 2C). These mutations are in the DNA-binding domain of p53 and are known to perturb its function as a tumor suppressor [16, 17]. Despite a non-functional p53, centrinone B treatment strongly repressed the growth of CSC-406 and POP-092 cancer colorectal organoids (~93% and 84% decrease in the average organoid area respectively) (Fig 2A and 2B) whereas the POP-112 line was relatively resistant to centrosome loss (only ~22% decrease in the average organoid area) (Fig 2A and 2B). To confirm that the centrinone-induced growth defect observed in CSC-406 and POP-092 lines was indeed p53-independent, we generated p53 null alleles in these lines using a lentiviral CRISPR-Cas9 system. p53 depletion efficiency was verified by western blot analysis (Fig 2D and 2E). Our results show that p53 knockout did not prevent the centrinone-induced growth defect in both CSC-406 and POP-092 lines confirming that this phenotype is p53-independent (Fig 2A and 2B). Since POP-112 was resistant to centrinone B, we then asked if centrosomes were readily lost in this line. Immunofluorescence staining for pericentrin, a centrosomal marker, confirmed that the centrinone B dose used (0.5 μM) led to efficient centrosome depletion (Fig 2F and 2G). We further carried out dose-dependent growth experiments in the POP-112 line. As shown in Fig 2H, 10 μM centrinone B was needed to induce a growth defect comparable to the extent of growth defect observed in CSC-406 and POP-092 achieved with only 0.5 μM centrinone B. Notably, centrosomes were lost in POP-112 at a centrinone B concentration 20x less that required to induce a growth defect suggesting a centrosome-independent growth arrest in response to centrinone B in this line.

Altogether, our results show that centrosome loss inhibits human normal colon organoid growth in a p53-dependent manner. However, cancer colorectal organoids exhibit different sensitivity to centrosome loss independently of p53, suggesting an additional regulatory mechanism.

## Mechanisms underlying centrinone sensitivity in cancer HCOs

Our data show that the centrinone-induced growth defect observed in CSC-406 and POP-092 cancer lines is p53-independent (Fig 2A and 2B). The three cancer organoid lines used were derived from different patients and are expected to carry different mutational signatures. To pinpoint the potential mechanisms involved in this growth defect, we first examined the mutation status of APC, K-Ras and SMAD4, which are the main colorectal cancer driver genes besides p53 [18]. Whole exome sequencing data (S1 File, Fig 3A) revealed a positive correlation between sensitivity to centrinone B and the presence of non-functional APC alleles. Indeed, the two centrinone-sensitive lines CSC-406 and POP-092 carried a non-functional APC mutation (nonsense Q1045* / Fs insertion P1594Afs*38 mutations for CSC-406 and nonsense G1499* mutation for POP-092) whereas the centrinone-resistant line, POP-112 possessed wild-type APC. Additionally, the POP112 line did not harbour mutations or deep amplifications or deletions in the WNT pathway members defined as colorectal cancer drivers APC, AXIN1, AXIN2, CTNNB1, GSK3B or RNF43, and only a shallow deletion in ZNRF3

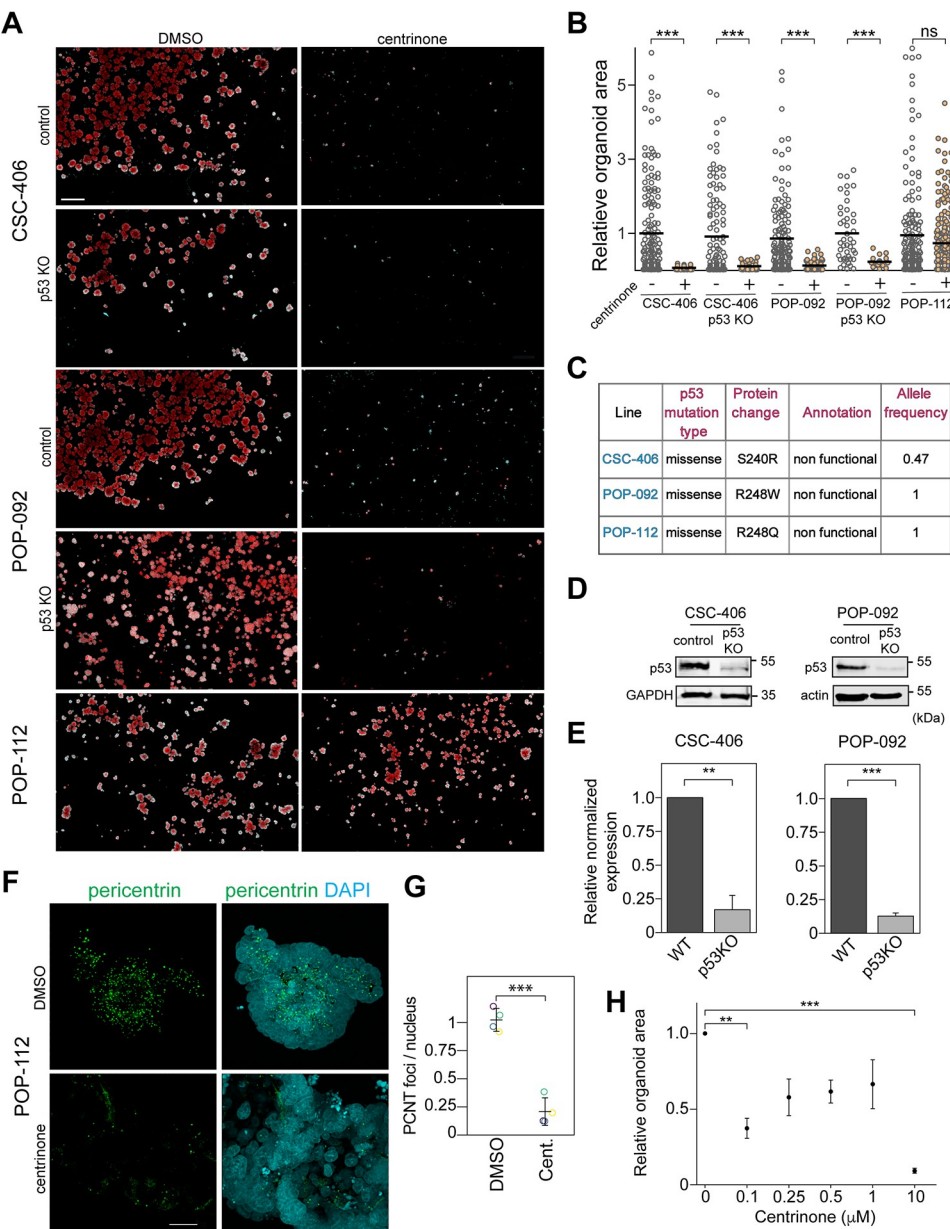

**Fig 2. Patient-derived cancer HCOs exhibit different sensitivity to centrosome loss independently of p53. A)** The indicated cancer HCO lines (control or p53 KO) were grown from single adult stem cells in the presence of DMSO or 0.5 μM centrinone B for 8 days. Organoids were then fixed, stained with DAPI (blue) and phalloidin (red) and imaged. Merged maximum intensity projection images are shown. **B)** The areas of individual organoids were quantified in the merged maximum intensity projection images from (A) and presented in the graph as relative values normalized to the respective DMSO control; every dot represents an organoid (n = 3, ***$P<0.001$, One-way ANOVA with Bonferroni post-hoc). **C)** p53 mutation status in the three cancer HCOs from the whole exome sequencing data. **D)** Western blot analysis of p53 and loading controls (GAPDH and actin) in lysates prepared from the indicated control and p53 KO cancer HCO lines. **E)** Quantification of (D). Protein levels from three independent experiments were quantified and presented in the graph (n = 3, **$P<0.01$, ***$P<0.001$). **F)** High-resolution maximum intensity projections of pericentrin (centrosome) and DNA (DAPI) immunofluorescence staining in POP-112 organoids treated with DMSO or 0.5 μM centrinone B for 8 days. **G)** The number of pericentrin foci and nuclei were automatically identified in z-stack sections spanning the entire organoid. The foci to nucleus ratio was determined for four organoids in each condition (***$P<0.001$). **H)** POP-112 organoids were grown from single adult stem cells in the presence of DMSO (0) or the indicated centrinone B concentrations for 8 days. Organoid areas were quantified as in (B). Relative average organoid area from three independent experiments is presented in the graph (n = 3, **$P<0.01$, ***$P<0.001$, all other treatments compared to DMSO are not significant, One-way ANOVA with Bonferroni post-hoc). Scale bars are (A) 500 μm and (F) 25 μm.

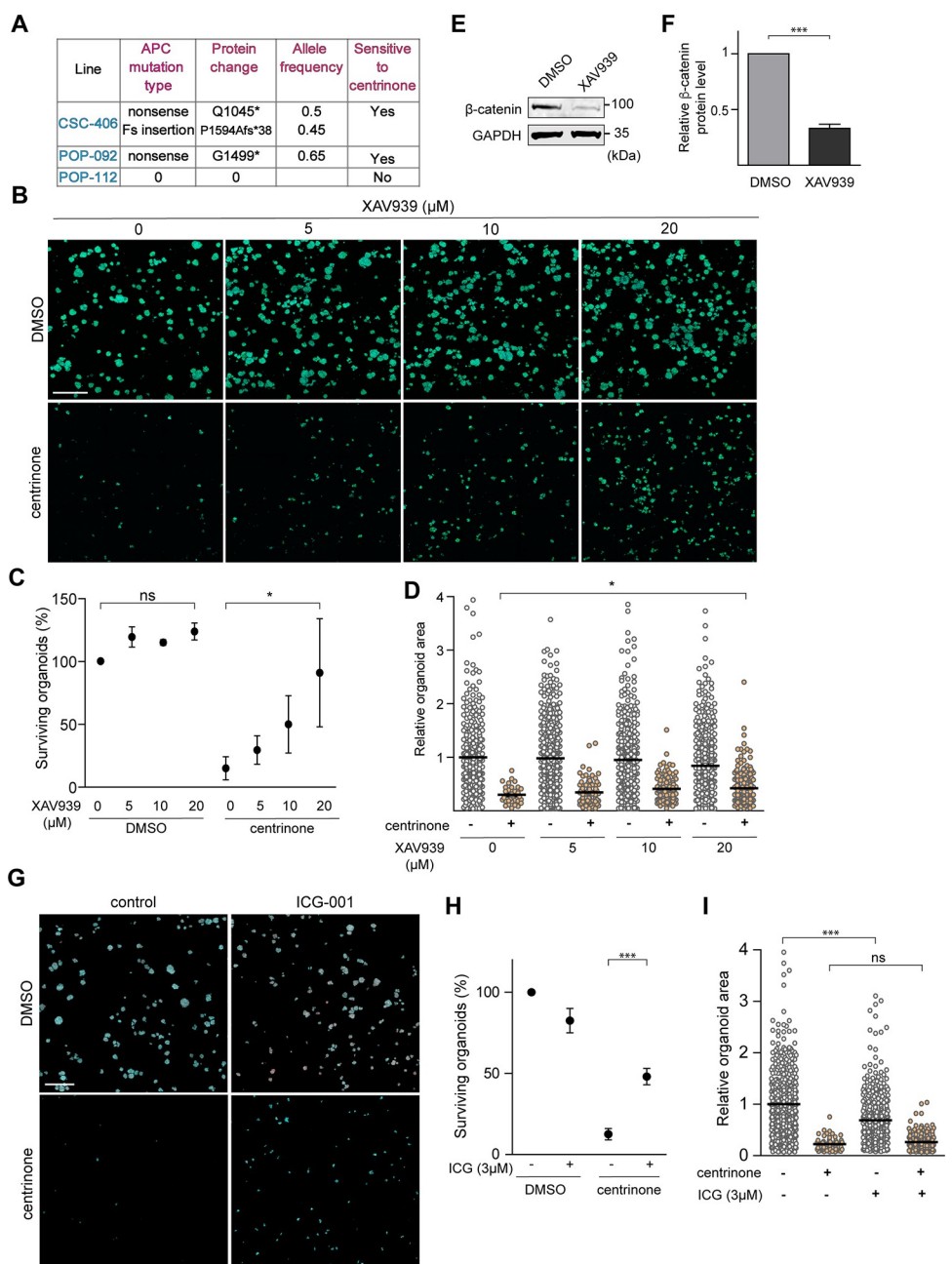

**Fig 3. Centrosome loss-induced organoid growth defect/death is β-catenin dependent in CSC-406 cancer line. A)** APC mutational status in the three cancer HCOs obtained via whole exome sequencing data. **B)** CSC-406 cancer organoids were grown from an equal number of single cells in the presence of DMSO or the indicated concentrations of XAV939 with DMSO or 0.5 μM centrinone B for 8 days. Organoids were then fixed, stained with DAPI (blue) and phalloidin (green) and imaged. Merged maximum intensity projections of representative images are shown. **C)** Percentage of surviving organoids in different conditions from (B) was quantified and presented in the graph (n = 3, *$P<0.05$, One-way ANOVA with Bonferroni post-hoc). **D)** The areas of individual organoids from B were quantified in the merged maximum intensity projection images and the relative values are presented in the graph, every dot represents an organoid (n = 3, *$P<0.05$, One-way ANOVA with Bonferroni post-hoc). **E)** CSC-406 cancer organoids were treated with DMSO or 20 μM XAV939 for three days, extracted from Matrigel and lysed. b-catenin and the loading control (GAPDH) protein levels were assessed using western blot analysis. The same set of Western blots was used to generate Fig 4E. **F)** Protein levels from three independent experiments were quantified and presented in the graph (n = 3, ***$P<0.001$, t-test). **G)** CSC-406 cancer organoids were grown from an equal number of single cells in the presence of DMSO or 3 μM ICG-001 and were treated with DMSO or 0.5 μM centrinone B for 8 days. Organoids were then fixed, stained with DAPI (blue) and phalloidin (red) and imaged. Merged maximum intensity projections of

representative images are shown. **H)** Percentage of surviving organoids in different conditions from (G) was quantified and presented in the graph (n = 3, ***$P<0.001$, One-way ANOVA with Bonferroni post-hoc). **I)** The areas of individual organoids from (G) were quantified in the merged maximum intensity projection images and the relative values are presented in the graph, every dot represents an organoid (n = 3, ***$P<0.001$, ns = non-significant, One-way ANOVA with Bonferroni post-hoc). Scale bars are (B and G) 500 μm.

(S1 File). These results suggested that centrinone-induced organoid growth defect might be mediated by hyperactive WNT signaling. To test this hypothesis, we inhibited the WNT signaling pathway downstream of APC using XAV939, a tankyrase inhibitor that increases the protein levels of axin, thereby promoting the degradation of β-catenin, the downstream effector of the canonical WNT pathway [19]. We treated CSC-406 organoids with increasing doses of XAV939 (0,5,10 and 20 μM) in the presence or absence of 0.5 mM centrinone B (Fig 3B–3D) and average organoid area and number of surviving organoids were then assessed. As shown in Fig 3C and 3D, centrinone B treatment resulted in a~ 85% decrease in surviving organoid number accompanied by a ~75% reduction in the organoid size. Remarkably, increasing doses of XAV939 rescued the growth rate and survival of centrinone B-treated organoids; a 20 μM XAV939 treatment significantly prevented ~75% centrinone-induced organoid death and increased the growth rate of the surviving organoids by ~40% (Fig 3C and 3D). This XAV939 concentration also efficiently inhibited the WNT pathway as it induced a ~67% decrease in β-catenin protein level as shown in western bot analysis (Fig 3E and 3F). To confirm our results, we targeted the WNT pathway using another inhibitor, Foscenvivint (ICG-001) that antagonizes β-catenin/TCF-mediated transcription by specifically binding to CREB-binding protein (CBP) [20]. We used a concentration of ICG-001 that was sufficient to decrease the steady state levels of the WNT pathway target survivin in MDA-MB-231 cells [21] (S1 Fig). Similarly to XAV939, ICG-001 partially prevented the centrinone-induced organoid death (Fig 3G and 3H) ~38% rescue of organoid survival rate) but the effect on growth rate was not significant (Fig 3I) possibly because 3 μM ICG-001 also reduced the DMSO-treated CSC-406 organoid growth (~30% reduction in average organoid area) (Fig 3I). Unlike XAV939 and ICG-001 that act downstream of the APC protein to inhibit the WNT pathway, using DKK1, a negative regulator of the WNT pathway that acts upstream of APC, did not rescue the growth/survival of centrinone B-treated CSC-406 organoids (S2A and S2B Fig).

To further investigate the role of the WNT pathway in the centrosome loss-induced organoid growth defect/death, we inhibited the WNT pathway in the other APC-mutant organoid line, POP-092 using the two inhibitors XAV939 and ICG-001 as we did with the CSC-406 line. In contrast to CSC-406, XAV939 treatment (5,10 and 20 μM) did not significantly prevent the centrinone-induced organoid growth defect/death (Fig 4A and 4B). To understand the difference between the two lines' response to XAV939 treatment, we determined β-catenin protein levels by western blot. We observed that the POP-092 line expressed ~90% less β-catenin protein than the CSC-406 line (Fig 4C and 4D). Moreover, XAV939 treatment reduced β-catenin protein levels by ~67% in CSC-406 organoids but surprisingly did not affect β-catenin protein levels in the POP-092 line (Fig 4C and 4D) probably explaining the absence of XAV939 effect on centrinone-induced organoid growth defect/death in this line. We asked whether the POP-092 line might be less sensitive to the drug and tested higher XAV939 concentrations (40 and 80 μM) and re-assessed β-catenin protein levels (Fig 4E). Surprisingly, β-catenin protein levels remained unaffected by these concentrations. To further explore the role of β-catenin in POP-092, we inhibited β-catenin/TCF mediated transcription with 3 μM ICG-001 in the presence or absence of centrinone B (Fig 4F). Similar to CSC-406, ICG-001 partially prevented the centrinone-induced organoid death in POP-092 (Fig 4F and 4G) but did not rescue the organoid growth rate (Fig 4H). We noted that 3 μM ICG-001 treatment also significantly decreased the

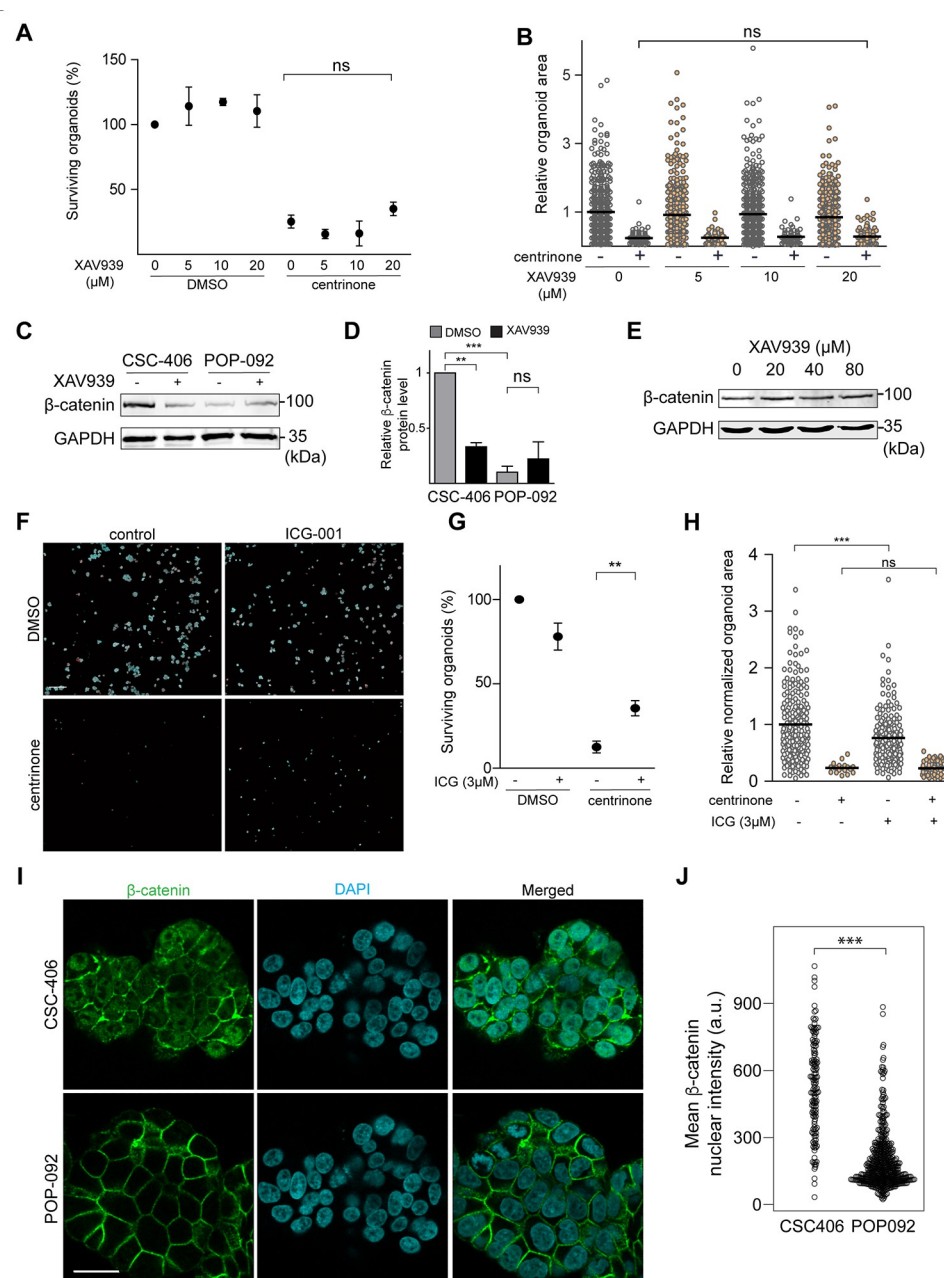

**Fig 4. β-catenin inhibition is essential for preventing centrosome loss-induced death of POP-092 cancer organoids. A)** POP-092 cancer HCOs were grown from an equal number of single adult stem cells in the presence of DMSO or the indicated concentrations of XAV939 and DMSO or 0.5 μM centrinone B for 8 days. Organoids were then fixed, stained with DAPI to label nuclei and phalloidin to label actin, and imaged. Merged maximum intensity projection images were used to quantify the percentage of surviving organoids in the different conditions (n = 3, ns: non-significant, One-way ANOVA with Bonferroni post-hoc). **B)** The areas of individual organoids from (A) were quantified in the merged maximum intensity projection images and the relative values are presented in the graph, every dot represents an organoid (n = 3, ns: non-significant, One-way ANOVA with Bonferroni post-hoc). **C)** CSC-406 and POP-092 cancer HCOs were treated with DMSO or 20 μM XAV939 for three days, extracted from Matrigel and lysed. b-catenin and the loading control (GAPDH) protein levels were assessed using western blot analysis. **D)** Protein levels from three independent experiments from (C) were quantified with Fiji and presented in the graph (n = 3, **P<0.01, ***P<0.001, ns: non-significant, One-way ANOVA with Bonferroni post-hoc). **E)** POP-092 cancer HCOs were treated with DMSO or the indicated XAV939 concentrations for three days, extracted from Matrigel and lysed. b-catenin and the loading control (GAPDH) protein levels were assessed using western blot analysis. A representative immunoblot is shown (n = 2). The same set of Western blots was used to generate Fig 3E–3F) POP-092

cancer HCOs were grown from an equal number of single cells in the presence of DMSO or 3 μM ICG-001 and were treated with DMSO or 0.5 μM centrinone B for 8 days. Organoids were then fixed and stained with DAPI to label DNA (blue) and phalloidin to label actin (Red). Representative maximum intensity projection images are shown. **G)** Percentage of surviving organoids in the different conditions tested in (F) was quantified (n = 3, **$P<0.01$, One-way ANOVA with Bonferroni post-hoc). **H)** Organoid areas from (F) were quantified in the merged maximum intensity projection images and the relative values are presented in the graph, every dot represents an organoid (n = 3, ***$P<0.001$, One-way ANOVA with Bonferroni post-hoc). **I)** Single focal plane images of β-catenin and DAPI immunofluorescence staining in fixed CSC-406 and POP-092 cancer HCOs. **J)** The relative intensity of b-catenin within nuclear objects identified using the DAPI channel was determined for four independent z-sections from each organoid presented in panel I. (n = 4, ***$P<0.001$, Mann-Whitney U test). Scale bars are (F) 500 μm and (I) 25 μm.

DMSO-treated POP-092 organoid growth (~25% reduction in the average organoid area) (Fig 4G). Finally, we compared β-catenin localization by immunofluorescence in the two lines. While β-catenin localized at the membrane in both lines, we found that the CSC-406 line exhibited much higher nuclear enrichment of the protein compared to POP-092 (Fig 4I and 4J).

## Discussion

Previous reports have shown that centrosome loss causes a p53-dependent growth arrest of normal human cells and that p53-mutant cancer cells continue to grow in the absence of centrosomes [12, 22]. Consistent with this, our results show that centrosome loss represses human normal colorectal organoid growth in a p53-dependent manner. It has been also shown that TRIM37 overexpression sensitizes breast tumors to centrosome loss [23]. However, our study reveals an unexpected β-catenin-mediated mechanism that controls centrosome loss-induced growth defect independently of p53 in cancer human colorectal organoids. Although p53 KO normal HCOs continued to grow without centrosomes, many abnormal nuclei phenotypes were observed including big multilobed nuclei and multi-nucleated cells most likely due to mitotic defects given the role of the centrosome in enhancing the mitotic spindle assembly efficiency [24, 25]. Despite all three cancer colorectal organoid lines carrying a non-functional mutant p53, only one line (POP-112) was resistant to centrosome loss as expected [12]. Although centrinone-resistant, this line was slightly more sensitive to low centrinone B concentrations (0.1 μM) than to the concentration needed to deplete centrosomes (0.5 μM, Fig 2H) most likely due to centrosome amplification induced by partial PLK4 inhibition. Indeed, our previous findings demonstrate that modulating PLK4 activity in cells leads to supernumerary centrosomes at low centrinone B concentrations and centrosome loss at higher concentrations [26]. Centrinone B is >1000-fold more selective against PLK4 than Aurora kinases AURKA and AURKB [12]. A high centrinone B concentration (10 μM) results likely in a cross-inhibition of AURKA and AURKB leading to organoid growth defect of the POP-112 line independently of the centrosome.

We provide evidence that centrosome loss-induced growth defect is p53-independent in APC-mutant cancer colorectal organoids since these organoids were highly sensitive to centrinone B treatment despite harbouring a non-functional p53 [16, 17]. It has been proposed that the S240R p53 mutant (mutation in CSC-406) retains some residual (10%) transcriptional activity [27]. Given that p53 depletion did not prevent the centrinone-induced growth defect, we assume that the latter is completely p53-independent in APC-mutant colorectal cancer.

Moreover, we demonstrate that the centrosome loss-induced growth defect/death of the APC-mutant cancer colorectal organoids tested is mediated by β-catenin, the downstream effector of the canonical WNT pathway. Indeed, decreasing β-catenin protein levels using XAV939 treatment significantly prevented the centrinone-induced organoid death and partially rescued the growth of CSC-406 organoids whereas, when β-catenin protein levels were unresponsive to XAV939 treatment in the POP-092 line, there was no significant effect of

XAV939 treatment on rescuing the centrosome loss-induced growth defect/death. XAV939 is a Tankyrase1/2 inhibitor that stabilizes Axin, a central scaffolding protein in the β-catenin destruction complex leading to β-catenin phosphorylation, ubiquitination, and degradation [19, 28]. It is not clear why XAV939 treatment in the POP-092 line does not affect β-catenin protein levels. Axin protein levels are regulated through Tankyrase-mediated PARsylation and ubiquitination targeting it for proteasomal degradation [19]. The unresponsiveness of the POP-092 line to XAV939 may be due to an altered signaling mechanism upstream of Axin (PARsylation and ubiquitination) or a failure to enhance the β-catenin destruction complex assembly downstream of Axin. Exploring Axin protein levels in response to XAV939 treatment will help address this question.

The role of β-catenin in mediating centrosome loss-induced cancer organoid death was confirmed in both APC-mutant lines (CSC-406 and POP-092) since the inhibition of β-catenin/TCF mediated transcription using ICG-001 partially prevented centrinone-induced death of both lines, although the effect on organoid growth rate was insignificant compared to XAV939, most probably due to ICG-001 cell toxicity effect [20]. Finding the balance between an effective and nontoxic ICG-001 concentration was challenging. This might explain the better organoid growth rescue obtained with XAV939 which is better tolerated due to the incomplete inhibition of β-catenin. Comparison of the APC-mutant cancer organoids shows that APC is more frequently mutated in CSC-406 (nonsense Q1045*, allele frequency 0.5 / Fs insertion P1594Afs*38, allele frequency 0.45) compared to POP-092 (nonsense G1499*, allele frequency 0.65) consistent with higher WNT pathway activity in CSC-406 that correlates with higher β-catenin protein compared to the POP-092 line. This may also explain why the CSC-406 line is slightly more sensitive to centrosome loss and responds better to β-catenin inhibition since it is more dependent on WNT/ β-catenin pathway compared to the POP-092 line.

Our study also highlights the genetic heterogeneity present in colorectal cancer [29]. Two organoids with APC mutations were sensitive to centrosome loss independent of p53 (CSC-406 and POP-092) while POP-112 was sensitive to centrosome loss, but in a p53-depedent manner. Interestingly, POP-112 did not harbour any loss of function mutations or deletions in APC, but rather had a shallow amplification at this locus (S1 File). Additionally, this line had a shallow deletion in the WNT pathway inhibitor ZNRF3 (S1 File). While loss of a single copy of ZNRF3 is associated with gonad development defects and increased WNT signalling [30], this might be suppressed by the shallow APC amplification. Although our sample size is small, we note that APC mutations are found in ~80% of sporadic and ~85% of familial cases of colorectal cancer, respectively while p53 is mutated in >80% of colorectal cancers [31]. Our observation that increased WNT signalling sensitizes colorectal cancers to centrosome loss even in the absence of p53 suggests that many, but not all, colorectal cancers might be treated therapeutically by targeting the centrosome. Biologically, it would be interesting to determine if increased WNT signalling even in resistant lines, such as POP-112, sensitizes these organoids to centrosome loss. It was previously shown that the 17q23 chromosomal amplification, harbouring TRIM37, promotes sensitivity to centrinone in breast cancer cells [23, 32]. Although TRIM37 is not amplified in the POP-112 line (S1 File), it is apparent that sensitivity to centrosome loss can be affected by the inactivation or amplification of multiple genes.

In addition to its role as a transcription factor, b-catenin has also been directly implicated in centrosomal functions where β-catenin localizes to the centriole linker in interphase cells and redistributes to the mitotic spindle during mitosis [33]. Additionally, β-catenin promotes the accumulation of the microtubule nucleating protein γ-tubulin at centrosomes and participates in centrosome disjunction [34, 35]. It is possible that excess β-catenin disrupts centrosome function rendering the organoids sensitive to centrosome loss. However, since ICG-001 binds directly to CBP to block its interaction with β-catenin, this drug should not directly

affect other β-catenin activities, thus we favour a model where β-catenin transcriptional activity is required for cancer organoid growth arrest after centrosome loss.

In summary, our study identifies WNT/ β-catenin as a new signaling pathway that mediates centrosome loss-induced growth defect and death in APC-mutant cancer colorectal organoids independently of the canonical p53 pathway. Therefore, targeting the centrosome may be a good strategy to combat colorectal cancer with non-functional p53 and hyperactive WNT/ β-catenin pathway.

## Materials and methods

### Reagents

Centrinone B (*Tocris Bioscience* #5690/10), Alexa Fluor™ 488 Phalloidin (Invitrogen #A12379), Alexa Fluor™ 546 Phalloidin (Invitrogen #A22283), DAPI (Invitrogen #21490), XAV939 (Cayman Chemical #13596–10), ICG-001 (Cayman Chemical #16257–5), Nutlin-3a (*Cayman Chemical* #10004372).

### Antibodies

p53 (Santa Cruz #SC126, WB: 0.2 μg/ml, IF: 0.8 μg/ml), CEP192 (Bethyl #A302-324A, IF: 4 μg/ml), pericentrin (Abcam #ab4448, IF: 1 μg/ml), GAPDH (Sigma #G9545, WB: 1 μg/ml), β-actin (Sigma #A5316, WB: 1 μg/ml), β-catenin (Santa Cruz # sc-7963, WB: 0.2 μg/ml, IF: 1 μg/ml), anti-mouse Alexa Fluor 488 (ThermoFisher # A-21202, IF: 0.5 μg/ml), anti-rabbit Alexa Fluor 488 (ThermoFisher # A-21206, IF: 0.5 μg/ml).

### Human organoid lines

Normal and cancer Human Colorectal Organoids (HCOs) were obtained with informed written patient consent from UHN Princess Margaret Living Biobank under REB 17–5519 (PMLB, Toronto, Canada). All studies with the HCOs described here were approved by the Mount Sinai Research Ethics Board (MSH REB Study # 18-0101-E). Three cancer organoid lines derived from archived tissues (retrospective study, 2017–2019) from three different patients: CSC-406 (female, 69 years old, colorectal adenocarcinoma), POP-092 (female, 46 years old, colorectal adenocarcinoma) and POP-112 (male, 74 years old, colorectal adenocarcinoma) were used in our study. The normal organoid line and CSC-406 cancer line were derived from the same patient. We had no access to information that could identify individual participants during or after data collection. To make the p53 KO normal and cancer lines, we produced lentiviruses in HEK293T cells using TLCV2 lentiviral vector (Addgene #87360) expressing a Tet-inducible CRISPR-Cas9 protein, in which we cloned a gRNA targeting the genomic p53 sequence: TATCTGAGCAGCGCTCATGG as described by the cloning protocol provided by Addgene (#87360). Organoids were broken up into single cells using a 40 min incubation (37˚c) with TrypLE™ express (Gibco™ #12605–028) followed by filtering through a 30 μM cell strainer (PluriSelect #43-50030-03). Single cells were then infected with the lentivirus in the complete growth media on a Matrigel layer. The day after, adult stem cells were attached to the Matrigel. Media was removed, another layer of Matrigel was added and fresh growth media containing puromycin (3 μg/ml) was supplemented. After 2 days, CAS9 protein expression was induced using 1μg/ml Tetracycline. After 3 days, p53 KO organoids were enriched by Nutlin-3a treatment (10 μM for 2 weeks) [14]. p53 Knockout efficiency was verified using TIDE analysis, immunofluorescence, and Western Blot.

## TIDE analysis

Genomic DNA was extracted from control and p53 KO organoid lines (PureLink™ Genomic DNA Mini Kit, Invitrogen™ #K182001) and a PCR was performed to amplify the genomic region targeted by the p53 guide using the primers: CTCAACAAGATGTTTTGCCAAC (Forward) and ACTCGGATAAGATGCTGAGGAG (Reverse). The PCR products were then Sanger sequenced using the Forward primer. Analysis of the two resulting raw sequencing files using the TIDE web tool (https://tide.nki.nl/) identifies the indels and their frequencies, giving a knockout efficiency score [15].

## Organoid culture

Normal and cancer Human Colorectal Organoids (HCOs) were obtained from UHN Princess Margaret Living Biobank (Toronto, Canada). HCOs were embedded in Matrigel domes and maintained in growth media composed of: Advanced DMEM/F-12 (Gibco™ #12634010), 100 U/ml penicillin/streptomycin (Gibco™ #15140–122), 10 mM HEPES (Gibco™ #15630–056), 2 mM GlutaMAX™ (Gibco™ #35050), 1.25 mM *N*-acetyl-cysteine (Sigma-Aldrich # A9165-5G), 1× B27 supplement (Gibco™ #17504–044), 10 nM gastrin (Sigma #G9145), 50 ng/ml mouse EGF (Gibco™ #PMG8041), 100 ng/ml mouse Noggin (Peprotech #250–38), 0.5 μM TGFb type I Receptor inhibitor A83-01 (Tocris #2939). In addition, Normal HCO media was supplemented with 40% WNT-3a conditioned media, 10% R-spondin conditioned media (UHN Princess Margaret Living Biobank, Toronto, Canada), 2.5 μM CHIR 99021 (Tocris #4423) and 10 μM p38 MAPK inhibitor, SB202190 (Sigma-Aldrich # S7067) while the cancer HCO media contained 10 μM of the rock inhibitor Y27632 (MedChem Express # HY-10583). Media was changed every 2 to 3 days and organoids were passaged every 7–9 days. Normal HCOs were broken up using enzymatic digestion using TrypLE™ express for 1–2 min followed by mechanical disruption (rigorous 10 times up and down pipetting) while only the enzymatic method (10 min TrypLE™ Express) was used for passaging the cancer HCOs. In experiments when starting organoid cultures (normal and cancer) from single cells is needed, a 30–45 min TrypLE™ Express incubation at 37˚c was performed, followed by cell filtering through a 30 μM cell strainer (pluriSelect #43-50030-03).

## Immunofluorescence

Organoids were embedded in Matrigel and grown on Nunc™ Lab-Tek™ II Chambered Coverglasses (Thermoscientific™ # 155360) for the indicated days. After cold PBS wash on ice (10 min), organoids were fixed with cold 4% paraformaldehyde (at 4˚c for 20–25 min). Cold PFA causes Matrigel meltdown allowing organoids to be fixed to the bottom of Chambered Coverglasses. After 1h permeabilization with 0.5% cold triton and 2-3h blocking with 1.5% BSA solution containing 0.1% triton, organoids were incubated with the primary antibodies in the blocking solution overnight at 4˚C. The day after, organoids were washed three times with PBS and incubated with secondary antibodies coupled with Alexa Fluor-488 at 0.5 μg/ml concentration for 2-3h. DNA was stained with DAPI and F-actin with Phalloidin (1/500). Organoids were PBS washed, then imaged in the imaging solution (0.7 mM N-acetyl cysteine, PH~7.4).

## Imaging and image analysis

Fixed and stained organoids were imaged in Nunc™ Lab-Tek™ II Chambered Coverglasses (Thermoscientific™ # 155360) using Nikon A1 HD25/A1R HD25 confocal microscope equipped with NIS-Elements controller. 20X and 40X water immersion objectives were used

to take z-stack images. To quantify organoid size and number, merged maximum intensity projections (DAPI and Phalloidin) of large images were processed in Fiji software as follows: image > type > 16-bit > adjust threshold > analyze particles. Fiji presents then the count and the size of all areas in the image. Each area represents an organoid.

## Western blotting

Organoids were extracted from Matrigel using TrypLE $^{TM}$ Express (20 min) and then lysed in lysis buffer (50 mM Tris-HCl, 1% NP-40, 10% glycerol, 140 mM NaCl, 5 mM $MgCl_2$, 20 mM NaF, 1 mM NaPPi and 1 mM orthovanadate, pH 7.4) complemented with protease inhibitor cocktail (Roche #11697498001). Total soluble proteins were measured using BCA Protein Assay Kit (Thermo Scientific #23227), then denatured in SDS sample buffer. Equal protein amounts were run on polyacrylamide gels and transferred onto PVDF membranes. After blocking, membranes were incubated with the specific primary antibodies in the blocking solution overnight. After TBS-Tween wash, membranes were incubated with the fluorescently labeled secondary antibodies and proteins were detected using Licor Odyssey CLx (LI-COR Biosciences—U.S). The digital images obtained were quantified using Fiji software.

## qPCR

Organoids were extracted from Matrigel using TrypLE $^{TM}$ Express (20 min) and mRNA extracted using a QIAGEN RNeasy Plus Mini Kit. Equal amounts of RNA (0.5 to 1 μg) was used to generate cDNA using the High-Capacity cDNA Reverse Transcription Kit (Thermo-Fisher). Resulting cDNA was used for qPCR using the following primers: RPLP0 forward 5′ – GAAACTCTGCATTCTCGCTTC 3′ and reverse 5′ – GGTGTAATCCGTCTCCACAG 3′; Survivin 5′ AGAACTGGCCCTTCTTGGAGG 3′ and reverse 5′ – CTTTTTATGTTCCTCTAT GGGGTC 3′.

## Statistical analysis

Statistical analysis was performed using a two-tailed t-test, one-way or two-way ANOVA analysis of variance followed by a Bonferroni's multiple comparison test or Dunnett post-hoc test using GraphPad Prism (version 8, https://www.graphpad.com/scientific-software/prism/).

## Supporting information

**S1 Fig. ICG-001 reduces the level of survivin mRNA in MDA-MB-231 cells. A)** MDA-MB-231 cells were treated with DMSO or 3 or 6 μM ICG-001 for 72 h before collecting cells for mRNA extraction. qPCR for surviving was performed on the subsequent cDNA using RPLP0 as a control. Relative survivin mRNA was normalized using RPLP0 and compared to levels in DMSO-treated cells. (n = 2, ***$P$<0.001, One-way ANOVA with Dunnett post-hoc using DMSO as control).
(TIF)

**S2 Fig. DKK1 does not rescue centrinone-induced growth arrest. A)** CSC-406 cancer HCOs were grown from an equal number of single adult stem cells in the presence of DMSO or the indicated concentrations of DKK1 and DMSO or 0.5 μM centrinone B for 8 days. Organoids were then fixed, stained with DAPI to label nuclei and phalloidin to label actin, and imaged. **B)** The areas of individual organoids from (A) were quantified in the merged maximum intensity projection images and the relative values are presented in the graph, every dot represents an organoid. (n = 1, ***$P$<0.001, Mann-Whitney U test).
(TIF)

**S1 File. Cancer organoid whole exome sequencing.**
(XLSX)

**S2 File. All raw data used for quantification.**
(XLSX)

**S1 Raw images. Unaltered original data for all Western blots.** Boxed regions indicate cropped area used for Figure preparation.
(PDF)

## Acknowledgments

We thank Mardi Fink from Dr. Jeff Wrana's lab at the Lunenfeld-Tanenbaum Research Institute for her help with establishing colorectal organoid culture in our lab.

## Author Contributions

**Conceptualization:** Mohamed Bourmoum, Laurence Pelletier.

**Data curation:** Mohamed Bourmoum.

**Formal analysis:** Mohamed Bourmoum.

**Funding acquisition:** Laurence Pelletier.

**Investigation:** Mohamed Bourmoum, Nikolina Radulovich, Amit Sharma, Johnny M. Tkach.

**Methodology:** Mohamed Bourmoum, Nikolina Radulovich.

**Project administration:** Mohamed Bourmoum, Laurence Pelletier.

**Resources:** Mohamed Bourmoum, Nikolina Radulovich, Ming-Sound Tsao, Laurence Pelletier.

**Software:** Mohamed Bourmoum, Nikolina Radulovich.

**Supervision:** Ming-Sound Tsao, Laurence Pelletier.

**Validation:** Mohamed Bourmoum.

**Visualization:** Mohamed Bourmoum, Amit Sharma, Johnny M. Tkach.

**Writing – original draft:** Mohamed Bourmoum.

**Writing – review & editing:** Mohamed Bourmoum, Amit Sharma, Johnny M. Tkach, Laurence Pelletier.

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
