## [Decision Letter · Decision Letter 0]

29 May 2023

PONE-D-23-09580β-catenin mediates growth defects induced by centrosome loss in APC mutant colorectal cancer independently of p53PLOS ONE

Dear Dr. Pelletier,

Thank you for submitting your manuscript to PLOS ONE. After careful consideration, we feel that it has merit but does not fully meet PLOS ONE’s publication criteria as it currently stands. Therefore, we invite you to submit a revised version of the manuscript that addresses the points raised during the review process. Please submit your revised manuscript by Jul 13 2023 11:59PM. If you will need more time than this to complete your revisions, please reply to this message or contact the journal office at plosone@plos.org. Please include the following items when submitting your revised manuscript:A rebuttal letter that responds to each point raised by the academic editor and reviewer(s). You should upload this letter as a separate file labeled 'Response to Reviewers'.A marked-up copy of your manuscript that highlights changes made to the original version. You should upload this as a separate file labeled 'Revised Manuscript with Track Changes'.An unmarked version of your revised paper without tracked changes. You should upload this as a separate file labeled 'Manuscript'.

We look forward to receiving your revised manuscript.

Kind regards,

Subhadip Mukhopadhyay, PhD

Academic Editor

PLOS ONE

Journal Requirements:

“This work was funded by grants from the Krembil Foundation and the CCSRI to LP.”

Reviewers' comments:

Reviewer's Responses to Questions

**Comments to the Author**

1. Is the manuscript technically sound, and do the data support the conclusions?

Reviewer #1: Yes

Reviewer #2: Yes

2. Has the statistical analysis been performed appropriately and rigorously? 

Reviewer #1: Yes

Reviewer #2: I Don't Know

3. Have the authors made all data underlying the findings in their manuscript fully available?

Reviewer #1: No

Reviewer #2: Yes

4. Is the manuscript presented in an intelligible fashion and written in standard English?

Reviewer #1: Yes

Reviewer #2: Yes

5. Review Comments to the Author

Reviewer #1: General

Authors do not comment on current literature regarding Wnt/β-catenin signaling and centrosomes. It would be useful to include for context in the introduction or discussion.

It would be valuable to repeat centrosome experiments with Wnt signaling inhibitors in normal organoids. A lingering question is if Wnt signaling plays a role in centrosome-mediated growth sensitivity in both normal and cancerous tissue, especially since p53 KO alone does not completely rescue centrinone-repressed organoid growth.

Figure 1

• It would be valuable to show p53 is not increasing cell survival through a centrosome-independent mechanism, especially because there is increased cell growth both with and without the addition of centrinone.

• 1C: CEP192 quantification should be included with these immunofluorescence images.

• 1E: Immunofluorescence images should be quantified, or western blots of p53 should be included with quantification to show knockout efficiency.

Figure 2

• POP112 is WT for mutations in APC, but does it have other mutations in Wnt pathway (i.e. β-catenin, GSK3β)?

• 2B: Legend suggests there are statistics, but they are not shown in the figure.

• 2C: Protein should be quantified.

• 2E: Loss of pericentrin should be quantified.

• 2F: Needs statistics.

Figure 3

• Validation of loss of Wnt signaling in ICG-001 treated cells is needed.

• It would be beneficial to show data with DKK1 in a supplemental figure, rather than stating “data not shown”.

• It is unusual that turning off Wnt pathway is increasing organoid growth, especially in the absence of centrinone. Rather, ICG decreasing organoid area as in 3H and 4G is what might be expected when Wnt pathway inhibitor is added to these cells. It might be helpful to speculate on this, as Wnt signaling is canonically known to increase cell proliferation.

• Similar to above, it is stated “data not shown” regarding Wnt inhibitors reducing organoid growth – is this a non-specific effect, or do you expect inhibiting Wnt signaling would stop organoid growth?

Figure 4

• Whole exome sequencing included in supplemental shows a β-catenin mutation in POP-092. Could this be related to loss of β-catenin protein? APC mutations are typically associated with increased β-catenin protein, so the results in 4C are unexpected.

• Since only one patient has mutant APC and high levels of Wnt/β-catenin signaling, it would be valuable to include other patients to validate your results. Currently, the manuscript is effectively working with a sample size of one. More specifically, it would be useful to repeat experiments with 1-2 mutant APC, β-catenin expressing organoid lines.

• 4H: Nuclear localization should be quantified.

Reviewer #2: [1] Overall comments

This manuscript describes a role of WNT signaling pathway in regulating centrosome loss-induced growth arrest in colorectal cancer. The finding is important, and the manuscript is well written.

[2] Minor Points

(1) It would be nice to quantify the images of β-catenin staining of Fig.4H.

(2) Figure Legend of Fig.1: scale bars should be (A) 500µm, (C) 10 µm, (E) 25 µm.

(3) Where are asterisks for Fig.2B?

(4) P-values are required for Fig.2F.

(5) Inconsistency: *** on Fig.3G but legend shows *; which one is correct?

(6) Which figure used one-way ANOVA or two-way ANOVA? Statements of statistical analysis in Figure Legend are required.

6. PLOS authors have the option to publish the peer review history of their article (what does this mean?). If published, this will include your full peer review and any attached files.

Reviewer #1: No

Reviewer #2: No

---

## [Author Response · Author response to Decision Letter 0]

13 Oct 2023

We thank the Reviewers for their thoughtful assessment of our manuscript and for their excellent suggestions. Below is our detailed point-by-point response to these comments and suggestions (original comments are in italics and our responses are in bold). The referee reports have been very helpful and we hope that the Reviewers will find the revised version of our manuscript suitable for publication in PLOS ONE.

Reviewer#1

1. Comments: Authors do not comment on current literature regarding Wnt/β-catenin signaling and centrosomes. It would be useful to include for context in the introduction or discussion.

It would be valuable to repeat centrosome experiments with Wnt signaling inhibitors in normal organoids. 

We thank the reviewer for suggesting we add this information. We now discuss the role of �-catenin function at the centrosome (Page 8, Paragraph 4) where we now include the following text “In addition to its role as a transcription factor, �-catenin has also been directly implicated in centrosomal functions where�� -catenin localizes to the centriole linker in interphase cells and redistributes to the mitotic spindle during mitosis [29]. Additionally, �-catenin promotes the accumulation of the microtubule nucleating protein �-tubulin at centrosomes and participates in centrosome disengagement [30, 31]. It is possible that excess �-catenin disrupts centrosome function rendering the organoids sensitive to centrosome loss. However, since ICG-001 binds directly to CBP to block its interaction with �-catenin, this drug should not directly affect other �-catenin activities, thus we favour a model where �-catenin transcriptional activity is required for cancer organoid growth arrest after centrosome loss.

Concerning the reviewer’s second point, since normal human colon organoid growth is strictly dependent on the WNT/�-catenin signaling pathway (Ootani A, et al) the organoids are grown in WNT-conditioned media containing the WNT agonist R-spondin. As such, we expect that organoids would not be viable in the presence of WNT inhibitors. This contrasts with cancer colon organoids that are viable in the absence of exogenous WNT activation.

2. Comments: A lingering question is if Wnt signaling plays a role in centrosome-mediated growth sensitivity in both normal and cancerous tissue, especially since p53 KO alone does not completely rescue centrinone-repressed organoid growth.

This comment is related to the second part of comment 1. Please see our response above. 

3. Comments: Figure 1

• It would be valuable to show that p53 is not increasing cell survival through a centrosome-independent mechanism, especially because there is increased cell growth both with and without the addition of centrinone

Indeed, in our control experiment, the loss of p53 alone resulted in an approximately 0.5-fold increase in organoid size. Since p53 is involved in multiple cell cycle checkpoints, loss of p53 function results in a faster growth rate (Brosh R, et al.), at the expense of genome instability, and could account for the larger size of these organoids. However, the loss of p53 resulted in a 5.5-fold increase in organoid size in the presence of centrinone B and subsequent centrosome loss. This still represents an approximate 10-fold stronger phenotype compared to the control. To formally address this, one could potentially inject mature centrosomes into the centrinone-treated organoids to determine if this suppresses the increase in organoid size after p53 loss. However, this experiment would be extremely technically challenging in any system, in particular in an organoid system and is beyond our lab’s current capabilities.

• 1C: CEP192 quantification should be included with these immunofluorescence images

We used automated image analysis to quantify the number of CEP192 foci and nuclei to determine a ‘foci to nucleus’ ratio for each organoid. We could not reliably quantify the CEP192 foci in centrinone-treated human normal organoids and this could be due to their small size and loss of viability upon extended centrinone B treatment. However, we provide quantification of foci in the p53 KO cells as a demonstration that centrinone B is depleting centrosomes as expected. This data is now included in the manuscript (Figure 1H) and discussed on page 4. 

• 1E: Immunofluorescence images should be quantified, or western blots of p53 should be included with quantification to show knockout efficiency.

We performed Western blot analysis for p53 in human normal organoids and their p53 KO counterparts. We now provide a representative Western blot and quantification data from three independent experiments in the manuscript (Figure 1E and F) on Page 4.

4. Figure 2

• POP112 is WT for mutations in APC, but does it have other mutations in the Wnt pathway (i.e. β-catenin, GSK3β)?

We surveyed our whole exome sequencing data for POP-112 for the main WNT pathway genes defined as colorectal cancer drivers (Bugter JM, et al) and did not find any mutations or deep amplification or deletion in APC, AXIN1, AXIN2, CTNNB1, GSK3� or RNF43. We did detect, however a shallow deletion in ZNRF3. Loss of a single copy of ZNRF3 is associated with gonad development defects (Harris A, et al.) However, POP-112 also harbours a shallow amplification of APC that might offset the effect of ZNRF3 loss on WNT signalling. These observations have been incorporated into the manuscript (Page 5, Paragraph 3 and Page 8, Paragraph 3).

• 2B: Legend suggests there are statistics, but they are not shown in the figure.

We apologize for this omission. The indications for statistical significance have been added to the figure.

• 2C: Protein should be quantified.

We repeated the Western blots for p53 comparing WT (control) and p53 KO organoids and quantified the relative abundance of p53. This data is now presented in Figures 2D and E and discussed on Page 5.

• 2E: Loss of pericentrin should be quantified

Similar to CEP192 in Figure 1, we used automated analysis to determine the number of pericentrin foci and nuclei within distinct sections for each organoid analyzed and determined a ‘foci to nucleus’ ratio for each organoid. These data are now presented in Figure 2 G and discussed on Page 5.

• 2F: Needs statistics.

We apologize for this omission. The indications for statistical significance have been added to the figure, which is now presented as Figure 2H.

5. Figure 3

• Validation of loss of Wnt signaling in ICG-001 treated cells is needed

We treated CSC-406 and POP-092 organoids with 3 and 6 �M ICG-001 for 72h and prepared cells for qPCR to detect survivin, a well-established target of the WNT pathway (Bao R, et al). However, we did not detect any changes in the levels of survivin mRNA. Since the WNT pathway activity is highest in a sub-population of cells within the entire colon organoid, such as Lgr5+ stem cells (Merenda A, et al), it is possible that our assay was not sensitive enough to detect these changes. The WNT pathway is active in the MDA-MB-231 breast cancer line so we repeated the experiment using these cells. As expected, we observed ~60% reduction in survivin mRNA after 72 h of ICG-001 treatment indicating that this compound is functioning normally. This data is now presented in the supplemental information (S1 fig) and discussed on Page 6. 

• It would be beneficial to show data with DKK1 in a supplemental figure, rather than stating “data not shown”

We agree that this data should be presented in a supplemental figure. We now include this data in Supplemental Figure 2A and B and discussed on Page 6.

• It is unusual that turning off Wnt pathway is increasing organoid growth, especially in the absence of centrinone. Rather, ICG decreasing organoid area as in 3H and 4G is what might be expected when Wnt pathway inhibitor is added to these cells. It might be helpful to speculate on this, as Wnt signaling is canonically known to increase cell proliferation

It is not clear which data the reviewer is referring to. Figure 3C shows a slight increase in organoid survival after XAV939 treatment but these changes are not significant and this has now been indicated in Figure 3C. Figure 3D indicates that the organoid area does not change after XAV939 treatment. We do, in general, observe a decrease in organoid survival and size after treatment with ICG-001, but not XAV939. Perhaps we see a noticeable effect on organoid survival and size after ICG-001 treatment since this inhibitor directly blocks �-catenin transcriptional activity while XAV939 lowers �-catenin abundance, but does not directly block its activity. 

• Similar to above, it is stated “data not shown” regarding Wnt inhibitors reducing organoid growth – is this a non-specific effect, or do you expect inhibiting Wnt signaling would stop organoid growth?

First, we have removed any ‘data not shown’ statements from the manuscript. Second, WNT signaling is required for colorectal organoid growth (Ootani A, et al). In normal organoids, WNT signaling is provided by adding exogenous Wnt3a from conditioned media and R-spondin to the growth media (Sato T, et al.) Even though cancer organoids provide their own WNT signaling due to mutations in the WNT pathway, they are still dependent on this pathway and thus both normal and cancer colorectal organoids are sensitive to WNT inhibitors. Indeed, we expect that organoid growth will be blocked in the presence of WNT pathway inhibition. In our work, we were careful to use sub-lethal concentrations of the WNT pathway inhibitors to decrease, but not abolish WNT signaling, so we could detect any survival or growth defects in the presence and absence of centrosomes.

6. Figure 4

• Whole exome sequencing included in supplemental shows a β-catenin mutation in POP-092. Could this be related to loss of β-catenin protein? APC mutations are typically associated with increased β-catenin protein, so the results in 4C are unexpected

We thank the reviewer for this observation. We would like to clarify that our exome sequencing data does not indicate that there is a mutation, but rather a shallow deletion in �-catenin. Although POP-092 expresses less �-catenin than CSC-406 this is not an appropriate comparison. We feel that a better comparison of relative �-catenin levels in POP-092 would be non-cancerous organoids from the same patient. Even if �-catenin levels in POP-092 are low, we think it is difficult to a priori determine the WNT pathway activation status in this line. We also noticed shallow deletions in CTNNBIP1 and DRAXIN, both �-catenin inhibitors that could compensate for lower �-catenin expression. Additionally, the destruction complex members Axin1 and Axin2 show shallow deletion and amplification, respectively, and might also influence �-catenin levels.

7.Since only one patient has mutant APC and high levels of Wnt/β-catenin signaling, it would be valuable to include other patients to validate your results. Currently, the manuscript is effectively working with a sample size of one. More specifically, it would be useful to repeat experiments with 1-2 mutant APC, β-catenin expressing organoid lines.

It is well established that colorectal cancer is a highly complex and molecularly heterogeneous disease, as reviewed by Fanelli and colleagues (Fanelli GN, et al.). We therefore want to be careful not to overreach with the interpretation and generality of our observed results. The reviewer is correct that two of colorectal cancer lines tested (CSC-406 and POP-092) were sensitive to centrinone while POP-112 was not. Unfortunately, these were the only three lines with the correct genotype within the very small cohort available to us. Our results do indicate, however, that a subset of colorectal cancer can display selective sensitivity to centrinone treatment and that this sensitivity correlates with WNT pathway activity. We have modified the title, abstract and the discussion to clearly indicate that this would only apply to a subset of colorectal cancers and that more work will be required to determine exactly how large that subset is. The concept of selective sensitivity to centrinone is not a novel one. Indeed, it was previously shown that the amplification of 17q23 that contains the TRIM37 gene promotes sensitivity to centrinone in breast cancer cells (Meitinger F, et al., Yeow YZ, et al.). Interestingly, TRIM37 was shown to promotes epithelial mesenchymal transition in colorectal cancer (Hu C-E, et al.). More work will be required to determine if such a mechanism also exists in colorectal cancer cells. This is now mentioned in the discussion on Page 8, paragraph 3.

4H: Nuclear localization should be quantified.

We quantified these images by analyzing four independent z-planes from each organoid. We used the DAPI channel to identify nuclei and measured the corresponding intensity of �-catenin within these objects. The results from this analysis is now presented in Figure 4J and discussed on Page 6, paragraph 2.

 

Reviewer#2 

Overall comments

This manuscript describes a role of WNT signaling pathway in regulating centrosome loss-induced growth arrest in colorectal cancer. The finding is important, and the manuscript is well written.

1. Minor Points

 It would be nice to quantify the images of β-catenin staining of Fig.4H

We agree that these data should be quantified. We did so by analyzing four independent z-planes from each organoid. We used the DAPI channel to identify nuclei and measured the corresponding intensity of �-catenin within these objects. The results from this analysis is now presented in Figure 4J.

2. Figure Legend of Fig.1: scale bars should be (A) 500µm, (C) 10 µm, (E) 25 µm

We have changed the figure legend accordingly.

3. Where are asterisks for Fig.2B?

We apologize for this omission. Asterisks to indicate statistical significance have been added.

4. P-values are required for Fig.2F.

We apologize for this omission and have added appropriate P-values (now Figure 2H)

5. Inconsistency: *** on Fig.3G but legend shows *; which one is correct?

Thank you for noticing this. The correct statistical difference should be indicated by ‘***’. We changed the figure legend for (now) Figure 3H to reflect this.

6. Which figure used one-way ANOVA or two-way ANOVA? Statements of statistical analysis in Figure Legend are required.

We apologize for this omission. The statistical test used for each comparison has now been included in each figure legend.

 

References

Ootani A, Li X, Sangiorgi E, Ho QT, Ueno H, Toda S, Sugihara H, Fujimoto K, Weissman IL, Capecchi MR, Kuo CJ. Sustained in vitro intestinal epithelial culture within a Wnt-dependent stem cell niche. Nat Med. 2009 Jun;15(6):701-6. doi: 10.1038/nm.1951. Epub 2009 Apr 27. PMID: 19398967; PMCID: PMC2919216.

Brosh R, Rotter V. When mutants gain new powers: news from the mutant p53 field. Nat Rev Cancer. 2009 Oct;9(10):701-13. doi: 10.1038/nrc2693. Epub 2009 Aug 20. PMID: 19693097.

Bugter JM, Fenderico N, Maurice MM. Mutations and mechanisms of WNT pathway tumour suppressors in cancer. Nat Rev Cancer. 2021 Jan;21(1):5-21. doi: 10.1038/s41568-020-00307-z. Epub 2020 Oct 23. Erratum in: Nat Rev Cancer. 2020 Nov 4;: PMID: 33097916.

Harris A, Siggers P, Corrochano S, Warr N, Sagar D, Grimes DT, Suzuki M, Burdine RD, Cong F, Koo BK, Clevers H, Stévant I, Nef S, Wells S, Brauner R, Ben Rhouma B, Belguith N, Eozenou C, Bignon-Topalovic J, Bashamboo A, McElreavey K, Greenfield A. ZNRF3 functions in mammalian sex determination by inhibiting canonical WNT signaling. Proc Natl Acad Sci U S A. 2018 May 22;115(21):5474-5479. doi: 10.1073/pnas.1801223115. Epub 2018 May 7. PMID: 29735715; PMCID: PMC6003506.

Bao R, Christova T, Song S, Angers S, Yan X, Attisano L. Inhibition of tankyrases induces Axin stabilization and blocks Wnt signalling in breast cancer cells. PLoS One. 2012;7(11):e48670. doi: 10.1371/journal.pone.0048670. Epub 2012 Nov 7. PMID: 23144924; PMCID: PMC3492487.

Merenda A, Fenderico N, Maurice MM. Wnt Signaling in 3D: Recent Advances in the Applications of Intestinal Organoids. Trends Cell Biol. 2020 Jan;30(1):60-73. doi: 10.1016/j.tcb.2019.10.003. Epub 2019 Nov 9. PMID: 31718893.

Sato T, Stange DE, Ferrante M, Vries RG, Van Es JH, Van den Brink S, Van Houdt WJ, Pronk A, Van Gorp J, Siersema PD, Clevers H. Long-term expansion of epithelial organoids from human colon, adenoma, adenocarcinoma, and Barrett's epithelium. Gastroenterology. 2011 Nov;141(5):1762-72. doi: 10.1053/j.gastro.2011.07.050. Epub 2011 Sep 2. PMID: 21889923.

Fanelli GN, Dal Pozzo CA, Depetris I, Schirripa M, Brignola S, Biason P, Balistreri M, Dal Santo L, Lonardi S, Munari G, Loupakis F, Fassan M. The heterogeneous clinical and pathological landscapes of metastatic Braf-mutated colorectal cancer. Cancer Cell Int. 2020 Jan 29;20:30. doi: 10.1186/s12935-020-1117-2. PMID: 32015690; PMCID: PMC6990491.

Meitinger F, Ohta M, Lee KY, Watanabe S, Davis RL, Anzola JV, Kabeche R, Jenkins DA, Shiau AK, Desai A, Oegema K. TRIM37 controls cancer-specific vulnerability to PLK4 inhibition. Nature. 2020 Sep;585(7825):440-446. doi: 10.1038/s41586-020-2710-1. Epub 2020 Sep 9. PMID: 32908304; PMCID: PMC7501188.

Yeow ZY, Lambrus BG, Marlow R, Zhan KH, Durin MA, Evans LT, Scott PM, Phan T, Park E, Ruiz LA, Moralli D, Knight EG, Badder LM, Novo D, Haider S, Green CM, Tutt ANJ, Lord CJ, Chapman JR, Holland AJ. Targeting TRIM37-driven centrosome dysfunction in 17q23-amplified breast cancer. Nature. 2020 Sep;585(7825):447-452. doi: 10.1038/s41586-020-2690-1. Epub 2020 Sep 9. PMID: 32908313; PMCID: PMC7597367.

Hu CE, Gan J. TRIM37 promotes epithelial mesenchymal transition in colorectal cancer. Mol Med Rep. 2017 Mar;15(3):1057-1062. doi: 10.3892/mmr.2017.6125. Epub 2017 Jan 16. PMID: 28098873; PMCID: PMC5367363.

---

## [Decision Letter · Decision Letter 1]

14 Nov 2023

β-catenin mediates growth defects induced by centrosome loss in a subset of APC mutant colorectal cancer independently of p53

PONE-D-23-09580R1

Dear Dr. Pelletier,

We’re pleased to inform you that your manuscript has been judged scientifically suitable for publication and will be formally accepted for publication once it meets all outstanding technical requirements.

Kind regards,

Subhadip Mukhopadhyay, PhD

Academic Editor

PLOS ONE

Additional Editor Comments (optional):

Reviewers' comments:

Reviewer's Responses to Questions

**Comments to the Author**

1. If the authors have adequately addressed your comments raised in a previous round of review and you feel that this manuscript is now acceptable for publication, you may indicate that here to bypass the “Comments to the Author” section, enter your conflict of interest statement in the “Confidential to Editor” section, and submit your "Accept" recommendation.

Reviewer #1: All comments have been addressed

Reviewer #2: All comments have been addressed

2. Is the manuscript technically sound, and do the data support the conclusions?

Reviewer #1: Yes

Reviewer #2: Yes

3. Has the statistical analysis been performed appropriately and rigorously? 

Reviewer #1: Yes

Reviewer #2: Yes

4. Have the authors made all data underlying the findings in their manuscript fully available?

Reviewer #1: Yes

Reviewer #2: Yes

5. Is the manuscript presented in an intelligible fashion and written in standard English?

Reviewer #1: Yes

Reviewer #2: Yes

6. Review Comments to the Author

Reviewer #1: Authors addressed all concerns and made appropriate revisions. The manuscript is now acceptable for publication.

Reviewer #2: (No Response)

7. PLOS authors have the option to publish the peer review history of their article (what does this mean?). If published, this will include your full peer review and any attached files.

Reviewer #1: No

Reviewer #2: No

---

## [Editor Report · Acceptance letter]

19 Jan 2024

PONE-D-23-09580R1 

PLOS ONE

Dear Dr. Pelletier, 

I'm pleased to inform you that your manuscript has been deemed suitable for publication in PLOS ONE. Congratulations! Your manuscript is now being handed over to our production team.

Kind regards, 

on behalf of

Dr. Subhadip Mukhopadhyay 

Academic Editor

PLOS ONE